# Identifying the Phenotypes of Diffuse Axonal Injury Following Traumatic Brain Injury

**DOI:** 10.3390/brainsci13111607

**Published:** 2023-11-20

**Authors:** Justin L. Krieg, Anna V. Leonard, Renée J. Turner, Frances Corrigan

**Affiliations:** Translational Neuropathology Laboratory, School of Biomedicine, Faculty of Health and Medical Sciences, The University of Adelaide, Adelaide 5000, Australia; justin.krieg@adelaide.edu.au (J.L.K.);

**Keywords:** traumatic brain injury, axonal injury, microtubules, neurofilament, spectrin

## Abstract

Diffuse axonal injury (DAI) is a significant feature of traumatic brain injury (TBI) across all injury severities and is driven by the primary mechanical insult and secondary biochemical injury phases. Axons comprise an outer cell membrane, the axolemma which is anchored to the cytoskeletal network with spectrin tetramers and actin rings. Neurofilaments act as space-filling structural polymers that surround the central core of microtubules, which facilitate axonal transport. TBI has differential effects on these cytoskeletal components, with axons in the same white matter tract showing a range of different cytoskeletal and axolemma alterations with different patterns of temporal evolution. These require different antibodies for detection in post-mortem tissue. Here, a comprehensive discussion of the evolution of axonal injury within different cytoskeletal elements is provided, alongside the most appropriate methods of detection and their temporal profiles. Accumulation of amyloid precursor protein (APP) as a result of disruption of axonal transport due to microtubule failure remains the most sensitive marker of axonal injury, both acutely and chronically. However, a subset of injured axons demonstrate different pathology, which cannot be detected via APP immunoreactivity, including degradation of spectrin and alterations in neurofilaments. Furthermore, recent work has highlighted the node of Ranvier and the axon initial segment as particularly vulnerable sites to axonal injury, with loss of sodium channels persisting beyond the acute phase post-injury in axons without APP pathology. Given the heterogenous response of axons to TBI, further characterization is required in the chronic phase to understand how axonal injury evolves temporally, which may help inform pharmacological interventions.

## 1. Introduction

Traumatic brain injury (TBI) occurs when a mechanical force is applied to the head. Current global statistics estimate that TBI affects between 800 and 1200/100,000 people, with a majority not requiring hospitalization [1]. One of the most significant consequences of a TBI is the development of traumatic axonal injury (TAI), which is known as diffuse axonal injury (DAI) when it is widespread. DAI is an injury feature in >80% of all motor vehicle accident-induced TBI cases and is consistently associated with worsened outcomes [2,3]. Importantly, the majority of axonal damage in TBI is not due to physical tearing of axons at the time of primary injury [4], but is caused by the initiation of a number of secondary injury processes which promote progressive axonal injury and eventual axonal swelling and detachment [5]. These secondary injury processes include an influx of calcium, degradation of the cytoskeleton due to activation of calpains and caspases, mitochondrial swelling, oxidative stress, and excitotoxicity, all of which have been comprehensively reviewed elsewhere [6,7,8,9,10]. Axonal injury persists chronically following even mild TBI, with axonal proteins found to be sensitive biomarkers of injury and predictive of long-term outcomes [11].

Despite the umbrella term of TAI or DAI, it is increasingly clear that axons, even within the same white matter tract, respond differently to the same mechanical insult, showing a range of cytoskeletal and axolemma alterations [4]. To understand injury evolution and develop targeted treatments, it is critical to understand how these axonal changes evolve following injury, which types of axonal alterations are likely to be present within the same axons, at what timepoint(s) post-injury, and how they can be detected. This review examines the specific cytoskeletal and axolemmal changes seen in post-mortem tissue following TBI and how they are currently identified.

## 2. Axon Structure

Axons are micron-thin tubular extensions generated by neuronal cells to transmit signals across large distances. Unlike the multiple short highly branched dendrites, which receive and integrate electrical synaptic input from thousands of neurons, only one axon is responsible for transmitting this integrated information as an action potential. To ensure information is transmitted properly, the axon has a unique cytoskeletal organization [12] with a central core of aligned microtubules, which are surrounded by neurofilaments, a membrane-associated cortex of cross-linked actin filaments, and an outer cell membrane, the axolemma [13]. Each of these elements is vulnerable to both the primary and secondary injury phase following TBI and can be identified via different markers following injury as discussed below.

## 3. Axolemma

Alterations to the physical appearance of the axolemma have been seen in ultrastructure studies in guinea pig optic nerve stretch [14], cat fluid percussion injury (FPI) [15], and rat weight drop [16]. Additionally, an ultrastructure study has also shown evidence of axolemma infolding from 6 to 30 h post-TBI in human tissue [17]. Specifically, axon membrane infolding and occasional disconnection are noted [17], which is likely representative of the degradation of the sub-axolemmal protein network that anchors the membrane to the cytoskeleton. Furthermore, the mechanical deformation associated with primary injury can create membrane pores, with an immediate increase in axolemma permeability [18,19,20,21,22,23,24], a process known as mechanoporation. Mechanoporation is principally assessed via the application of cell-impermeable dyes, with the intra-axonal presence of dye indicative of increased axolemma permeability [18,19,20,21,22,23,24], demonstrated in a severity-dependent manner. For example, in primary cortical neurons, severe stretch (10 s^−1^, 0.3 magnitude) caused immediate uptake of all tested dyes (380–150,000 Da) in up to 60% of neurons, whereas a milder stretch (1 s^−1^, 0.1 magnitude) was associated with minimal dye uptake, regardless of dye size [25]. Similarly, Smith et al. found that a 60% strain rate led to no uptake of a 570 Da dye, whereas a modest uptake was noted at a 75% strain rate [26]. Initial mechanoporation in cell culture models appears to be brief, lasting < 10 min following the initial mechanical exposure [25,27], but there may be a secondary phase that is linked to secondary injury processes, with dye entry noted at 60 min post-stretch in a 3D co-culture model [28].

In vivo models show a similar injury severity effect with moderate–severe [18,19,20,21,22,23], but not mild injury [20,29] associated with acute axonal dye uptake, evident within 5 min of injury. Like in vitro models, variable uptake was seen within bundles of fibers, with some axons exhibiting dye uptake, but not others, supporting the differential responses of axons in the same anatomical location to injury [18,19,20,21,22,23]. Infusion of dye intraventricularly at 5 h post-injury in a cat FPI model still identified positive axons [19], and similarly, in a rodent weight-drop model, 20% of all dye-positive neurons only contained dyes administered after injury at 8 h post-injury [30]. These studies indicate that beyond the primary mechanical insult, secondary processes continue to disrupt the axonal membrane, at least in the acute phase post-injury.

However, mechanoporation is not required for ionic fluxes to occur across the axolemma. It is well established that TBI leads to an immediate increase in intracellular calcium levels [31,32]. In vitro studies of dynamic stretch within unmyelinated axons show a massive influx of calcium immediately after injury [33,34,35,36], which is relative to the strain applied [35]. Increased intracellular calcium then persists for at least 24 h post-stretch, albeit at much lower levels [33]. Similarly, in vivo studies using autoradiography for ^45^Ca found significant accumulation post-lateral FPI up to 48 h post-injury [37]. Notably, with a milder stretch injury, axonal dye uptake was not present, and with simultaneous blockage of NMDA and AMPA receptors, and voltage-gated sodium and calcium channels, calcium influx had also not occurred, suggesting that mechanoporation was not significant at this injury severity [36]. This was not the case with a more severe stretch, where pharmacological blockade was no longer effective, indicating that mechanoporation had occurred [36]. Thus, calcium may enter the axon via a number of different pathways post-injury including activation of NMDA and AMPA receptors by unregulated glutamate release [10] and direct mechanical activation of voltage-gated sodium channels. Patch clamp studies show that pressure, bending, and expansion of the axonal membrane cause an increased conductance in these sodium channels [38]. The overall effect of an influx of sodium is depolarization, which then leads to the activation of voltage-gated calcium channels [34]. Increasing intracellular sodium also causes the Na^+^/Ca^2+^ exchanger to operate in reverse, further facilitating calcium influx [39]. Calcium may also enter the intracellular axonal space from intra-axonal calcium stores like the mitochondria [40,41,42], with pharmacological blockade of mitochondrial transition pores with cyclosporin A also attenuating intra-axonal calcium accumulation in cultured neurons following stretch injury [42].

Regardless of the initial mechanism, calcium influx promotes the activation of proteolytic enzymes including calpains and caspases-1, -3, -14, -17, and -18 [9]. Calpains exist as a heterodimer with a large catalytic subunit and a small regulatory subunit [43]. Calcium binding produces a conformational change to create an active catalytic unit that cleaves structural proteins, receptors, and ion channels [6]. Calcium also induces the opening of mitochondrial membrane transition pores and the release of cytochrome c, leading to caspase activation [44]. Axonal calpain activity is elevated within minutes of axonal stretch [31,43]. Furthermore, with calpain [43] and caspase-3 [45] proteolysis of the axonal structural protein, αII-spectrin is evident within 15–30 min following diffuse injury in rats. Calpain activation also causes persistent changes in sodium channels located within the axolemma [33,34,46,47]. Decreases in the immunoreactivity of the sodium channel Nav1.2 are evident within 20 min in unmyelinated axons in in vitro models of stretch injury [33,34,46]. Similarly, decreases in NaV1.6 immunoreactivity, only found on myelinated axons, persisted to 72 h in a rotational pig model of mild TBI, returning to sham levels by 14 days post-injury [47]. Specific regions of the sodium channel may be particularly vulnerable to calpain cleavage. The α-subunit, which forms the ion conduction pore, contains four transmembrane domains (I, II, III, and IV), which are connected through intracellular linkers [48]. By 20 min post-stretch, the III–IV intracellular linking region had significantly decreased on western blot and immunohistochemical analyses [33]. As proteolysis of this III–IV intracellular linking region prevents inactivation of the channel, loss of this region may permit persistent leakage of sodium into the neuron [49]. Calpain-mediated proteolysis was also noted within the I–II [33,46], II–III [46], and IV [33] transmembrane regions, observed as decreased expression of the full protein and appearance of smaller fragments, with a subset of these fragments remaining within the axolemma [46]. Given that the channel fragments remain in the axolemma and are no longer responsive to normal regulatory signals, this is thought to similarly permit ongoing sodium influx. Persistent depolarization activates voltage-gated calcium channels, with the subsequent calcium influx being a potential cause of ongoing axonal dysfunction and degeneration [33,34,47]. However, to date, sodium channel pathology has not been investigated beyond 14 days post-injury, so its role in the chronic phase post-TBI is not known.

## 4. Sub-Axolemmal Proteins

Directly underneath the axolemma, a membrane skeleton consisting of ring-like actin filaments connected by spectrin tetramers is located [50]. Each spectrin tetramer is formed by the association of two heterodimers comprising an αII and a β chain, with the β chain either the II or IV isoform, depending on axonal location [51]. The actin–spectrin complex is then anchored to the axolemma through various proteins including ankyrin, NMDA receptors, and Na^+^/H^+^ exchangers [52]. This structure is key for protecting axons from normal mechanical forces associated with movement, with β-spectrin-deficient *Caenorhabditis elegans* exhibiting fragile axons that break with movement [53]. Indeed, both computational and cell culture modeling have shown that spectrins can dynamically fold and unfold with tension due to their highly crosslinked nature, acting like a shock absorber [13,54].

The actin–spectrin skeleton is vulnerable to TBI. In an in vitro axonal injury model, axonal swellings within 50 s of injury developed where there was an increased spacing of the actin–spectrin rings in response to the mechanical force [32]. Furthermore, both α and β-spectrin are vulnerable to proteolysis by calpain and caspase-3, with the αII subunit producing 145/150 kDa and 120 kDa fragments, respectively [55]. Western blots targeting the αII-spectrin subunit showed an increase in both the 145/150 kDa and 120 kDa proteolytic fragments following a controlled cortical impact (CCI) injury in rats, indicating both calpain and caspase-3 activity [56]. The peak in these breakdown products is between 24 and 48 h [56,57], with up to 30-fold increases observed within the ipsilateral cortex [56]. These spectrin breakdown products are also detected in the CSF of rats [58] and humans [59] following TBI. Clinically, a different temporal profile is seen in calpain- compared to caspase-3-mediated spectrin cleavage, with the peak of calpain-mediated breakdown products at 1 day compared to 7 days for caspase-mediated breakdown products [59]. As these spectrin breakdown products are also found in the serum, they have been found to be predictive biomarkers of both cognitive decline and post-concussion syndrome in TBI uncomplicated by hemorrhage [60]. However, one inherent limitation of these methods is that α-spectrin is not only located in axons but also in other cell types like astrocytes. Indeed, spectrin breakdown products have been found within astrocytes in double labeling experiments [61]. Hence, the presence of spectrin breakdown products, without further validation, may not necessarily indicate axonal injury.

As such, immunohistochemical studies allow visualization of the αII-spectrin breakdown product within axons, with antibodies specific to target sites within the C-terminus (Ab39) or N-terminus (Ab37, Ab38, or spectrin N-terminal fragment [SNTF]) of the spectrin–calpain proteolysis site [62]. These epitopes are inaccessible within intact spectrin protein but become exposed upon proteolytic degradation to enable specific detection of axonal injury (Figure 1A) [62]. Calpain-mediated spectrin degradation is seen early following injury, with scattered Ab38-positive axons within the pyramidal tracts and medial lemnisci from 15 min post-injury, increasing to 2 h following diffuse weight-drop TBI in rats [63]. Work in stretch models of injury suggests that the appearance of axons with spectrin proteolysis changes over time, with Ab38 immunoreactivity evident within morphologically normal non-swollen axons from 20 to 30 min, but in swollen degenerating axons at 4 days [31]. Similarly, in a mild rotational pig model of TBI, SNTF immunoreactivity within axons was noted by 6 h, despite these axons appearing intact without notable varicosities or swellings, but then later evolved to beading within SNTF-positive axons at 48–72 h (See Johnson et al. for staining reference histology images) [64,65]. SNTF appears to predominantly be an early marker of injury. When sections were double labeled with the current gold standard marker of axonal transport disruption, amyloid precursor protein (APP), alongside SNTF, 44% of injured axons were detected with only SNTF at 6 h post-injury compared to 19–24% at 48–72 h post-injury [64]. In comparison, APP detected 51% of injured axons at 6 h, increasing to 75–79% at 48–72 h, with minimal overlap in SNTF and APP immunostaining suggesting that cytoskeletal elements can be differentially affected in injured axons [64]. Indeed, in a sheep model of mild TBI axonal injury at 4 h post-injury could only reliably be detected with SNTF, not APP labeling [66]. However, to date, whether spectrin degradation continues more chronically following TBI has yet to be reported.

Furthermore, whilst α-spectrin degradation has been extensively characterized, β-spectrins have not been widely studied. There is evidence of calpain- and caspase-mediated breakdown products of βII-spectrin following CCI injury from 2 h to 7 days, with a peak at 48 h post-injury [67]. These authors argue degeneration of this subunit has more functional implications due to disruptions in binding with ankyrin-G, compared to αII-spectrin, which does not bind to ankyrin-G [67]; however it must be noted that any disruption to the spectrin lattice has wider implications for the integrity of the axon given their importance in withstanding normal mechanical loads [13].

## 5. Microtubules

The central core of the axon is comprised of ~10–100 microtubules per cross-sectional view [68]. These microtubules facilitate axonal transport to and from the synapse, providing tracks used by molecular motors like dynein and kinesin for long-distance transport of proteins, synaptic vesicles, mitochondria, and other organelles [7]. Microtubules are assembled from polymerization of dimers of α- and β-tubulin, with 13 of these protofilaments then assembled around a hollow core [69]. Neighboring microtubules are crosslinked via microtubule-associated proteins, like tau [70].

Microtubules are the stiffest structural component of the axon [71], such that dynamic stretch results in a rapid rupture, triggering a gradual depolymerization and eventual loss of microtubules. Secondary injury processes like calpain activation further destabilize microtubules by hydrolyzing tubulin and microtubule-associated proteins, as well as directly promoting further depolymerization [72,73]. Indeed, post-injury reductions in microtubules have been shown via electron microscopy within the brainstem of cats post-FPI [19], in the optic nerve fibers of guinea pigs following stretch injury [74,75], and in myelinated axons of the human cortex following TBI [76]. In support of this, levels of the key structural protein alpha-tubulin are decreased by 30 min post-injury, with this decline persisting up to 3 days following weight-drop TBI [16].

However, microtubule vulnerability following TBI appears to be heterogeneous, as not all microtubules within an axon break in response to an insult. For example, an in vitro stretch model study reported that only around a third of microtubules were broken in specific regions along the axons [77], with tau playing a key role in microtubule rupture by inducing high stress at its binding sites under dynamic mechanical loads [78]. These breaks produce undulations within the axon, which progress to form multiple isolated axonal swellings or varicosities, representing regions of impaired axonal transport at the site of microtubule disruption (see Tang-Schomer et al. for staining reference histology images) [77]. Indeed, in an in vitro stretch model, 85% of axonal varicosities co-localized with local microtubule discontinuities [79]. Continued accumulation of transport products can lead to the development of axonal bulbs, seen as discrete swellings within axons [80]. It should be noted that although these were traditionally viewed as terminal bulbs, signifying disconnection of the axon, this may not always be the case and may simply reflect the limitations of the thin sections evaluated. In CLARITY-cleared tissue, axon tracing through 100 μm deep sections found that nearly 30% of these bulbs at 24 h following focal CCI injury were connected varicosities, with this decreasing to 10% of axonal bulbs at 1 month post-injury [81]. These findings suggest that even in the chronic post-injury phase, axonal bulbs may still reflect potentially reversible pathology.

To date, assessment of microtubule function post-injury has been predominantly assessed indirectly via the detection of the accumulation of axonally transported products. As previously mentioned, accumulation of APP, which undergoes fast axonal transport, is the current gold standard for the detection of axonal injury (Figure 1B) [4]. Clinical reports describe APP axonal pathology developing within 30 min of severe injury [82,83] and peaking within the first few days [84,85], with predominant involvement of the brainstem, internal capsule, thalamus, corpus callosum, and parasagittal white matter [84]. Current grading of diffuse axonal injury is based on the extent of APP pathology with Grade 1 exhibiting APP axonal damage only within the white matter of the cerebral hemispheres and grey/white matter interfaces, with involvement of the corpus callosum and brainstem in Grade 3 [86]. Interruption to axonal transport appears to be an ongoing injury process in TBI, with APP axonal pathology persisting at low levels chronically, at least in a subset of patients. Examination of the corpus callosum following moderate–severe TBI in those that survived greater than one year found APP-positive axonal pathology in ~60% of cases [87], whilst investigation of mild TBI found APP axonal pathology up to 6 months post-injury [88,89]. The appearance of the axonal injury changes over time with early APP axonal pathology consisting of long and tortuous varicosities, with isolated or small clusters of axonal bulbs in a single directional plane, which progresses to isolated and granular axonal bulbs [87].

Pre-clinical models replicate these clinical findings. APP-positive axonal injury is evident within hours following injury, peaking at around 24 h with some evidence of chronic pathology dependent on the injury model, brain region, and timepoint examined [90,91,92,93,94]. For example, moderate–severe injury induced by the Marmarou weight-drop model in rats produces maximal APP axonal pathology at 24 h in regions including the pyramidal tracts, corpus callosum, optic tract, cingulum, and fornix, with APP pathology still present at lower levels in these regions at 14 days following injury [90,91]. Unlike these white matter tracts, within grey matter regions like the thalamus and cortex, the peak of APP axonal injury may be delayed, although still developing within the first week [93]. Examination of chronic timepoints has found the presence of scattered APP-positive axonal profiles within the corpus callosum at 12–24 months following a single mild closed head injury [95,96]. More severe injury, like the focal CCI model, produces more chronic APP axonal pathology, detected from one month to one year post-injury in white matter and grey matter structures like the striatum and thalamus [97].

Comparable findings have been shown in gyrencephalic models of TBI. At 24 h following diffuse injury in ferrets, APP accumulation within axons was detected in a severity-dependent manner in the corpus callosum, fornix, internal capsule, brainstem, thalamus, and cortical white matter, with axonal injury most prominent at the midline within the grey/white matter interfaces and the periventricular regions [98]. In a pig rotational model of mild TBI, APP-positive pathology was similarly detected within the subcortical, periventricular, and midline structures, increasing from 6 h to 48–72 h following injury, the latest timepoint examined [64]. Following severe injury, APP-positive axonal bulbs were still present at 6 months, although reduced by 58–77% compared to acute timepoints [99].

Accumulation of APP has thus proven to be an excellent marker of axonal pathology, demonstrating graded increases in injury severity whilst still persisting at lower levels to chronic timepoints. However, beyond axonal transport disruption, there may be other avenues to investigate the fate of microtubules post-injury. Recent work has shown that microtubules can activate different neurodevelopmental programs post-injury that determine their fate. Following lateral FP injury, a subset of axons underwent post-translational modifications with excessive tubulin tyrosination and phosphorylation of tau, thereby promoting degeneration, whilst others had increased expression of the developmental microtubule-associated protein (MAP) MAP6 and maintenance of acetylation associated with microtubule stabilization and repair [47]. In support of this, MAP6 was found to be differentially expressed in the thalamus of blast-exposed compared to sham mice at one month post-injury [100]. Future work should therefore investigate the temporal evolution of these microtubule markers chronically post-injury to determine whether they continue to be associated with axonal fate and thus could be used as adjunct markers of axonal injury.

## 6. Neurofilaments

The final cytoskeletal element, neurofilaments are a type of intermediate filament that lie beneath the actin and spectrin cortex and surround the microtubules, acting as space-filling polymers [101]. Neurofilaments accumulate in axons during development to expand axon caliber, increasing axonal conduction velocity, and are thus predominantly found in large axons [102]. There are three neurofilament isoforms, namely light (NF-L; 68 kDa), medium (NF-M; 168 kDa), and heavy (NF-H; 200 kDa), with each consisting of a central α-helical rod region, a short variable head domain, and a tail that differs in length for each isoform [103]. The rod domain is important for co-assembly with other neurofilaments to form heteropolymeric filaments that run parallel along the length of the axon, with frequent cross-bridges between neighboring filaments (Figure 1C). The ratio of NFH:NFM:NFL within these filaments in a human adult is 2:3:7 [104], with the more abundant NFL forming the backbone and NF-M and NF-H located more peripherally, as their longer tails form sidearms that protrude from the filament core [105]. Indeed, compared to 142 residues in the NF-L tail, NF-M has ~514 and NF-H around 613 residues. The long NF-H and NF-M tail domains also contain phosphorylation sites within repeated serine-lysine proline (KSP) sequences, 51 sites in NF-H and 7 in NF-M, meaning that NF-H is more heavily phosphorylated [106]. Phosphorylation straightens, aligns, and bundles neurofilaments whilst extending the sidearms [107]. Sidearm extension promotes cross-bridge formation among neighboring filament bundles, alongside other cytoskeletal elements [108]. Phosphorylation increases the negative charge on these sidearms [109], with the repulsive negative charge between neighboring filament bundles acting to increase inter-filament spacing, thereby increasing axon diameter [110,111]. Of note, despite having a slightly shorter tail and fewer phosphorylation sites, the NF-M sidearm shows more lateral extension than NF-H with phosphorylation [112] and is thus thought to be the most important determinant of axon caliber [113].

The mechanical insult causes almost immediate effects on the neurofilament structure. Within 5 min of central FPI in cats, there is a significant decrease in interfilament spacing, with density analysis showing an increase in neurofilament packing [19]. Other injury-induced alterations to neurofilaments include misalignment [17,20] and accumulation of neurofilaments in ring- [114] or bulb-like [31] structures. These accumulations lack microtubules and contain a dense core or organelles surrounded by a whorl (for ring) or dense bulb of neurofilaments [17]. Neurofilament alterations appear to be injury severity-dependent. In an in vitro stretch model, mild strain caused increased neurofilament staining via SMI-312 (a marker of phosphorylated NF-M and NF-H) and marked development of ring-like neurofilament structures, whereas with greater strain, increased NF immunoreactivity was less apparent, with axons now distorted and disorganized and eventually undergoing secondary axotomy [114]. Given the antibody used, the loss of staining could reflect either dephosphorylation or loss of the protein itself. It should also be noted that axons demonstrating neurofilament compaction can recover, with an in vitro compression model finding 50% of compacted axons recovered within 1 day, and a further 10% by 7 days [115].

Nevertheless, loss of the sidearms themselves may drive neurofilament pathology, with deletion of the tail domain of both NF-H and NF-M resulting in a highly disorganized neurofilament cytoskeleton with minimal cross-bridging [116] and degradation of NF-L and the tail-less NF-M and NF-H subunits [117]. Dephosphorylation of neurofilament sidearms by calcineurin, another calcium-activated enzyme, also promotes neurofilament compaction by reducing the repulsive negative charge on the sidearms. Furthermore, both NF-M and NF-H are highly sensitive to proteases, such as calpain when the tail domain is dephosphorylated [118], causing neurofilament degradation.

Evaluation of the time course of neurofilament axonal pathology found that the earliest timepoint at which NF-L immunostaining could detect axonal injury was 6 h post-injury in a pig model of rotational TBI [119], consistent with clinical reports [17,120]. NF-L-positive axonal injury was also detected earlier than antibodies specific to NF-M sidearms (NN18) or NF-H sidearms (N52) [119,121], with faint staining at 24 h that became more prominent at 3 days and persisted out to 10 days following injury [119] (Figure 1C). At the 7 day timepoint, a previous comparison of various neurofilament antibodies found that antibodies directed to NF-L (NR4) and NF-H (N52) detected the most axonal injury, compared to NF-M (NN18) and phosphorylated specific NF-H (SMI-31) and non-phosphorylated specific NF-H (SMI-32) [122]. In a cat model of FPI, NF-L axonal injury was then decreased markedly between the 7 day and 14 day timepoints [123]. In both studies, the neurofilament axonal pathology had the same distribution as that reported for APP, located predominantly at anatomical borders like the grey/white matter interface and periventricular region, in addition to major white matter tracts [122,123], but neurofilament pathology was much less extensive than that detected with APP [64].

Few studies have investigated neurofilament pathology chronically post-injury. Doust et al. used SMI-34, which detects normal phosphorylated NF-H and found persistent axonal injury with bulbs, blebbing, thickening, and undulations, with 2–3× more axonal injury in the somatosensory barrel cortex than in the corpus callosum [124]. In this study, animals were injured with the lateral FPI model at different ages (17 days, 35 days, 2, 4, and 6 months) and formalin-fixed at 10 months of age, which equated to ~4–9 months post-injury. All rats showed comparable levels of axonal injury regardless of when injury had occurred [124]. Similarly, in a pig model of rotational TBI, N52, which recognizes both phosphorylated and non-phosphorylated forms of NF-H, still detected swollen axonal bulbs at 6 months post-injury, although this was considerably reduced compared to that observed acutely at 3–7 days post-injury [98]. This may be injury dependent, with Iliff et al. [125] only detecting sparse SMI-34 immunoreactive varicosities in the cortex and corpus callosum 28 days post-mild closed head impact and Yu et al. [126] finding no SMI-34 axonal pathology 4 months post-mild TBI. Nonetheless, like axonal transport defects, it appears that neurofilament pathology continues into the chronic post-injury phase in at least some types of TBI.

Histological examination of the effects of injury on neurofilaments is not limited to antibodies that target the normal protein, but also antigens within neurofilaments that are suggestive of pathology. For example, RMO-14, -44, -48, and -194 bind to antigens on the rod core of NF-M [127,128], which is only exposed following disruption of the sidearms and is thus suggestive of neurofilament compaction (see Stone et al. for histological reference images of neurofilament compaction [94]) [128]. Compared to targets of the NF-M sidearms [119,121], NF-M rod antibodies detect axonal pathology within 15 min of diffuse weight-drop TBI in rats [18], visualized as thin and elongated axonal profiles [18,94,129]. Axonal profiles increase in size to 24 h post-injury, where they reveal multiple vacuoles [129]. RMO-14 axonal pathology shows different peaks depending on the brain region examined. Following diffuse TBI in rats, RMO-14 immunoreactivity peaked at 3 h in the corticospinal tract compared to 24 h in the medial lemniscus, which has larger caliber axons with more neurofilaments [129]. In the gyrencephalic pig model, RMO-14 pathology was minimal at 6 h, peaked at 48 h, and had decreased by 72 h post-injury and was much less extensive than APP pathology at all timepoints [64,130]. Furthermore, although present within the same locations, there is minimal overlap between RMO-14 and APP pathology within individual axons in rats, pigs, and clinical TBI, suggesting that neurofilament compaction and axonal transport are separate injury processes [64,94,129]. This differs from spectrin proteolysis within axons, with early pathology (<2 h post-injury) showing almost complete co-localization of RMO-14 and Ab38 in a rat diffuse injury model [63]. However, by 6 h in a pig model of TBI, only ~50% of RMO-14-positive axons co-localized with SNTF, reducing to ~20% by 72 h, suggesting that there may be a divergence in the evolution of these two types of pathology over time [64]. Chronically, minimal RMO-14 pathology was found in the medial lemniscus and corticospinal tract at 30 days post-injury, despite APP pathology still being present [131]. No studies have looked beyond this timepoint, with further work needed to characterize the pattern of immunoreactivity long term following injury.

Another neurofilament marker suggestive of pathology is SMI-32, which targets non-phosphorylated NF-H, although its efficacy appears to be species-specific. For example, SMI-32 showed only faint staining in uninjured mice, with injury severity increasing in SMI-32-positive axons within the cortex and hippocampus from 24 h to 7 days post-TBI [132,133,134] and increased staining with optic nerve stretch injury peaking at 4 days post-stretch [31]. In contrast, SMI-32 immunoreactivity in pigs is higher in uninjured animals than that seen in mice. Indeed, SMI-32 did not readily identify axons with altered morphology following injury, even in regions where RMO-14 and NF-L detected axonal injury [64]. Similarly, in clinical cases, SMI-32 did not reveal extensive pathology, in part because of the high degree of baseline immunoreactivity [64]. The neurofilament protein shows some differences between mice and humans, with mice for example having more phosphorylation sites on NF-H (51 vs. 45) [109], which may account for why the antibody produces different staining patterns.

Novel NF-L antibodies that are neurodegeneration-specific in immunohistochemical investigations have recently been identified. The two NF-L antibodies currently used in the SIMOA NF-L assay are Clones 2.1 and 47.3, also known as UD1 and UD2, and bind to unique locations within the rod domain. Shaw et al. were able to demonstrate that both UD1 and UD2 only bound to injured axons in a contusion spinal cord injury model, with no staining in uninjured tissue, with speculation that the epitopic region must be inaccessible in assembled neurofilaments and revealed during degeneration [135]. Future work could investigate the temporal profile of axonal injury in these markers compared to other neurofilament subtypes, particularly given the recent interest in NF-L as a sensitive serum biomarker of both injury and recovery [11,136,137].

Given the varying targets of different NF antibodies and their ability to bind to both the normal protein and injury-specific targets, employing a range of different antibodies may provide a more comprehensive picture of neurofilament pathology post-injury. This is particularly important given early work showing different peaks of pathology depending on the antibody target. Furthermore, consideration should be given to the species and whether the antibody of interest is optimal.

## 7. Axon Initial Segment and Nodes of Ranvier

The above discussion does not differentiate specifically between the cytoskeletal structure of unmyelinated and myelinated axons. Fundamental to the function of myelinated axons are discrete unmyelinated sites that facilitate the propagation of the action potential. The axon initial segment (AIS) is located at the proximal region of the axon and serves as the site of action potential initiation, while nodes of Ranvier are small gaps (~1 μm long) between adjacent myelin sheaths, which allow rapid propagation of the action potential through saltatory conduction [138]. To facilitate this function, the AIS and nodes of Ranvier are characterized by high concentrations of voltage-gated ion channels, particularly the sodium channel NaV1.6 [48]. Ion channels and other membrane proteins are anchored to the plasma membrane by the scaffolding protein ankyrin-G, which, together with βIV-spectrin, provides a link to the underlying actin cytoskeleton [51]. The diameter of the axon also decreases at the node of Ranvier [139], which is accomplished by a decrease in neurofilament number [140,141]. For example, in a study by Price et al., the 82% reduction in axonal caliber was linked to a decrease in neurofilament number by 76% at the node of Ranvier [142] with no change in microtubule number [140,141]. Flanking the nodal region are the paranodes, where myelin loops are anchored to the axon through connections between glia (Nfasc155) and the axon (Caspr and contactin) to create the unmyelinated node of Ranvier [143]. These cell adhesion molecules are anchored by different classes of cytoskeletal proteins, with the paranode region characterized by ankyrin-B, Protein 4.1B, and βII-spectrin [50].

Notably, these unmyelinated regions of myelinated axons may be particularly vulnerable to axonal injury, as they experience larger strains than the rest of the axon [144]. Indeed, the majority of axotomy was shown to arise either at or near the AIS in a mouse model of central FPI [145], with contraction of the distal end of the AIS, measured via staining for ankyrin-G, identified within cortical pyramidal neurons [146]. Alterations in the AIS were associated with a reduction in action potential acceleration and reduced peak amplitude, indicating that these changes had effects on the functioning of the axons [146].

The nodes of Ranvier are equally vulnerable, with the formation of nodal blebbing within 15 min of injury identified via electron microscopy in both models of optic nerve stretch [14,74,147] and rotational TBI in non-human primates [74]. Microtubule loss is also particularly pronounced at the nodes of Ranvier, where microtubules outnumber neurofilaments due to the axonal constriction [74]. More sophisticated analysis has found noticeable disruption of the nodal regions even in mild TBI [47]. Following mild rotational acceleration in pigs, levels of NaV1.6 were decreased up to 3 days post-injury, whilst void nodes, where a paired Caspr domain indicative of a paranodal region flanks a node without NaV1.6 signal, and heminodes, where nodes of Ranvier only had one paranodal region, increased up to two weeks post-injury (Figure 1D) [47]. Similarly, in a mouse concussive model, Marion et al. found a shortening of the overall length of the paranode–node–paranode region, with an increase in the number of heminodes up to six weeks post-injury [148]. Following midline FPI, the total number of NaV1.6 nodes in the corpus callosum was also decreased, with an increase in heminodes [149]. Disruption in the nodal region is linked to alterations in the underlying cytoskeleton, with diffusion of key nodal cytoskeletal proteins βIV-spectrin and Ankyrin-G into the paranode identified [47]. Levels of ankyrin-G within the cerebellum have been found to decrease expression at 2 and 7 days following central FPI injury, with a return to sham level by 30 days [150], with mild FPI increasing ankyrin-G lysis products within the corpus callosum [151]. Notably, axons can show nodal pathology without accumulation of APP, with this seen in mice [149], pigs [47], and clinical cases of more severe TBI [47], indicating that nodal pathology can occur independently of axon transport defects and appears to persist for longer [47]. Given that this pathology may provide widespread disruption of axon potential propagation, it could play an important role in network dysfunction and loss of white matter integrity post-TBI. Thus, examination of key nodal and paranodal cytoskeletal and membrane proteins including Caspr, Ankyrin-G and Ankyrin-B, contactin, and NaV proteins could give important insights into the evolution of axonal injury post-TBI.

## 8. Co-Existence of Different Phenotypes of Axonal Injury

The previous sections outlined how TBI affects the axolemma and cytoskeletal elements individually. It appears that these different axonal elements are differentially affected by TBI. Neighboring axons within the same tract exposed to the same mechanical loading can develop different axonal injury phenotypes indicated by axolemma mechanoporation [18,20,30], APP accumulation [90,91,92,93,94], neurofilament alterations [18,94,129], calpain-mediated spectrin degradation [31,63,64,65], or sodium channel pathology [47,148,151], with varying degrees of overlap [20,22,63,64,94,129].

Indeed, mechanoporation sufficient to allow dye influx (3–40 Da) does not seem to be associated with either calpain-mediated spectrin degradation [30] or APP accumulation [22] in the acute phase following TBI. Axons with dye influx appear to rapidly degenerate, showing dramatic infolding, disconnection, and vacuolization from 30 min to 3 h post-injury. These axons show a thin elongated appearance as opposed to the bulbs that form with APP accumulation [22]. In contrast, neurofilament pathology can be identified in these axons, with evidence of neurofilament compaction within 5 min, and co-localization of HRP staining with NF-M rod antibodies up to 6 h following injury in a moderate–severe cat FPI model [18]. Neurofilament pathology is however not reliant on mechanoporation, with mild injury not sufficient to lead to intra-axonal horseradish peroxidase accumulation in axons still associated with neurofilament disarray and misalignment, with increased NF-L immunoreactivity [20].

Beyond the axolemma, differences in cytoskeletal pathology are also seen in different axons within the same tracts. At 6 h post-injury, a pig model of mild TBI found that the prominent axonal pathologies at the periventricular interface were accumulation of APP and calpain-mediated spectrin breakdown, with neurofilament pathology less prominent as detected via NF-68 (for NF-L) and RMO-14. Minimal overlap was seen between APP and SNTF immunolabelling, although with more severe clinical injury, SNTF/APP co-labelling was more prominent, suggesting that this may be injury severity dependent [64]. Although most SNTF-positive axons did not co-localize with neurofilament markers, in the small population of axons positive for either RMO-14 or NF-68 in the pig mild TBI model, ~50% co-localized with SNTF, which is unsurprising given that calpain can similarly break down neurofilaments. In contrast, APP and neurofilament pathology appear to predominantly occur in separate populations in the first 24 h post-injury [64,94,129]. At 30 min to 6 h following diffuse weight-drop TBI in rats, APP and RMO-14 label distinct axonal phenotypes, with APP confined to axonal swellings and RMO-14 to elongated axonal segments, with minimal overlap [94]. The degree of co-localization may also depend on the caliber of the axons examined, with the smaller axons within the corticospinal tract showing overlap in <1% of axons at 24 h compared to 27% within the larger axons of the medial lemniscus, which contain more neurofilaments [129].

After 24 h post-injury, axonal transport disruption, as detected by APP, is the most prominent form of axonal injury [64,84,85,90,91,92,93,94], with 5–6× more APP-positive pathology compared to SNTF, NF-68, or RMO-14 (Figure 2) [64,129]. Minimal overlap between axonal pathologies is observed. By 72 h post-injury, even axons showing neurofilament and spectrin pathology had diverged, with <20% of RMO-14-positive axons now co-labeling with SNTF [64]. Sodium channel pathology has emerged as a newer indicator of axonal injury, and intriguingly, although neurofilament, spectrin, and axonal transport disruption appear to co-exist within the same white matter tracts, if within different neurons [64,94,129], sodium channel changes appeared to be more widely distributed across the white matter compared to APP-immunoreactive swellings [47]. In addition, whilst other axonal pathologies peak within the first few days following injury, sodium channel and node of Ranvier pathological changes appeared to progress dynamically to at least two weeks post-concussion [47], suggesting a more enduring alteration in axons post-injury which could underlie ongoing symptoms. To date, the evolution of different axonal injury phenotypes chronically has received little attention, with further research needed in this area, specifically co-localizing markers for different axonal phenotypes with quantification.

## 9. Conclusions

Injury to axons is a complex process, highly dependent on axon caliber, structure, and force direction. Given that the axonal population responds in a heterogenous manner to the initial mechanical force, it is vital to use a variety of axonal injury markers examining the response of all the cytoskeletal elements to capture the full extent of axonal disruption, with this yet to occur in the chronic phase post-injury. Examination of the AIS and the nodal region post-injury has also emerged as key to understanding the evolution of symptoms following TBI, given that disruption to these regions may affect axon potential propagation. Although nodal pathology was first highlighted in the 1990s, a comprehensive examination of the evolution of pathology to the chronic phases post-injury incorporating the key cytoskeletal elements in the paranodal and nodal regions has yet to be conducted. A key component will be probing the colocalization of markers to understand how the injury response in different cytoskeletal elements does or does not influence other structures within the axon and the time course over when this occurs. Different types of axonal pathology may require different interventions and different treatments. Furthermore, given that axonal pathology is known to continue into the chronic phase post-injury, there may be opportunities to intervene even in the remote stage post-TBI.

## Figures and Tables

**Figure 1 brainsci-13-01607-f001:**
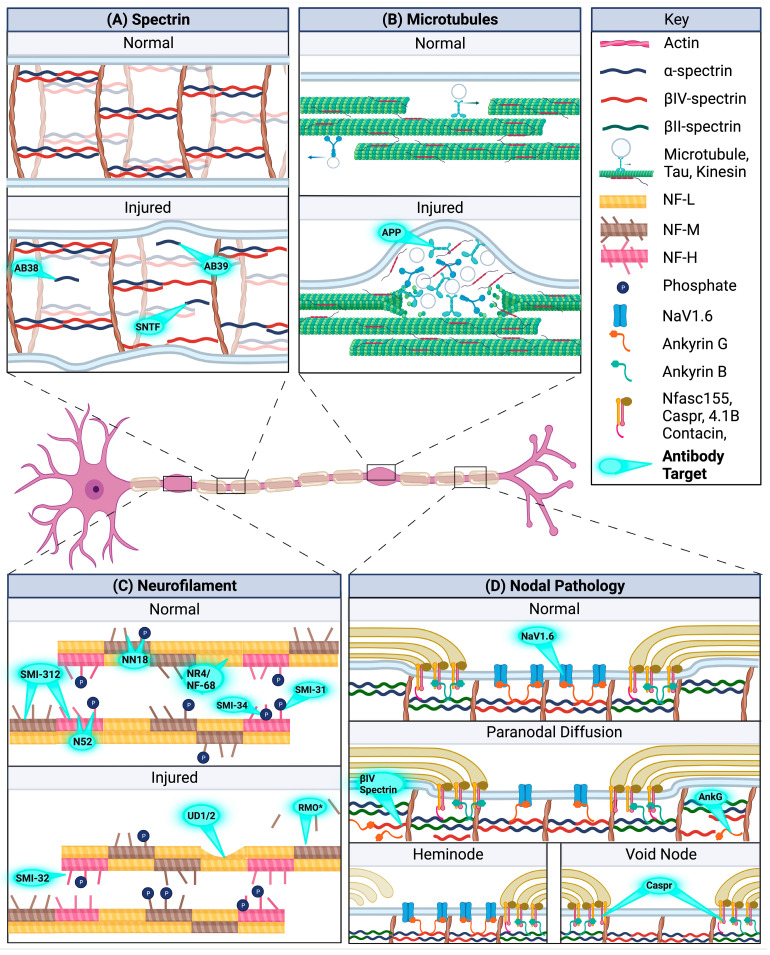
Disruption of different cytoskeletal elements can be visualized with different antibody markers. (**A**) α-spectrin proteolytic breakdown can be observed through antibodies to target sites of the calpain proteolysis site within the C-terminus (Ab39) or N-terminus (Ab38 or SNTF) [62]. (**B**) Microtubule disruption is typically examined via accumulation of fast axonally transported proteins, with APP as the gold standard. (**C**) Neurofilament pathology can be examined via a number of different antibodies that target both normal neurofilament proteins and those revealed only after injury. (**D**) Within myelinated axons, the nodes of Ranvier may be uniquely vulnerable to injury, with loss of NaV1.6 from the nodal region, or Caspr from the paranode with loss of diffusion of the nodal cytoskeletal proteins βIV-spectrin and ankyrin-G into the paranodal space. This can be visualized as heminodes or void nodes depending on the cytoskeletal pathology. Note RMO* refers to RMO-14, -44, -48, and -194. Created with Biorender.com.

**Figure 2 brainsci-13-01607-f002:**
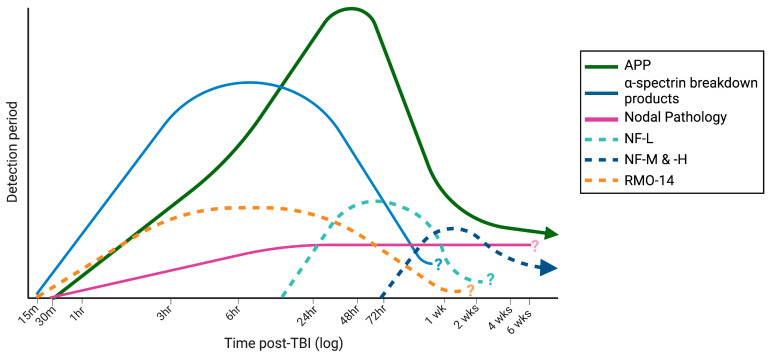
Temporal profile of cytoskeletal damage markers. Lines indicate the detection period for markers based on the current literature, as well as the anticipated peaks in pathology. APP pathology peaks around 24 h with a small amount of pathology present chronically. Calpain-mediated cleavage of α-spectrin peaks earlier at 6 h and then diminishes over time, although chronic pathology has not been examined. Neurofilament pathology, detected by RMO antibodies, peaks earlier than markers of the intact neurofilament proteins, with NF-L pathology evident earlier than NFM and NF-H with evidence that, at least in more severe TBI, neurofilament pathology may persist chronically. Nodal pathology develops early and appears to persist chronically, at least to six weeks post-injury, the latest timepoint examined. Question marks denote pathologies that have not been assessed chronically. Created with Biorender.com.

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
