# Peer review of "Identifying the Phenotypes of Diffuse Axonal Injury Following Traumatic Brain Injury"

_brainsci, 2023, doi:10.3390/brainsci13111607_

Round 1
Reviewer 1 Report
Comments and Suggestions for Authors
The paper delves into the mechanisms and evolution of Diffuse Axonal Injury (DAI), a notable aspect of Traumatic Brain Injury (TBI). It describes the structure of axons and how TBI impacts their cytoskeletal components differently, even within the same white matter tract. The discussion includes various detection methods for post-mortem analysis of axonal injury, emphasizing the significance of amyloid precursor protein (APP) accumulation due to microtubule failure as a sensitive marker. However, it notes that a subset of injured axons shows distinct pathologies, undetectable by APP immunoreactivity, like spectrin degradation and neurofilament alterations. Recent studies highlighting the vulnerability of the node of Ranvier and axon initial segment to injury, and persistent loss of sodium channels post-injury in axons without APP pathology are also discussed. The paper suggests that further examination of axonal injury's temporal evolution in the chronic phase post-TBI could aid in devising pharmacological interventions.
The work is well supported, and the evidence is laid out quite clearly. It would be of great interest to the readers of Brain Sciences.
Author Response
We thank the reviewer for their examination of the paper, with no revisions suggested.
Reviewer 2 Report
Comments and Suggestions for Authors
This manuscript provides an excellent overview of the complex degenerative changes associated with axonal proteins following TBI.
I have provide a few suggestions that the authors may wish to consider.
Line 91; Change 570Da to 570 Da. For constancy consider having space before Da and kDa throughout manuscript.
114-116: Sentence “Notably, with a milder stretch injury, mechanoporation sufficient to allow axonal dye uptake was not required for calcium influx, as calcium entry was prevented with simultaneous block-age of NMDA and AMPA receptors, alongside voltage-gated sodium and calcium channels [36].”
Could the wording for the first half of this sentence be improved/reworded, as I found it a bit confusing.
Paragraph beginning line 132, mentions calpains and caspases in opening sentence, but then goes on to only discuss calpains. Is it worth mentioning something about the consequences of caspase activation with respect to axonal injury following TBI.
Figure 1A: Is AB37 antibody target location correct (i.e., at N-terminal or should this be the AB39 anitibody) as indicated in Fig legend. Also, in text and Fig legend lower case “ab” is used and in Figure upper case “AB” is used.
Line 229: correct indentation at start of line.
Line 428: Does RMO stand for anything???
Check Lines 431 - 432: E.g., Sentence beginning ”Compare to…”.
Reviewer 3 Report
Comments and Suggestions for Authors
The paper reviews the phenotypes of diffuse axonal injury following traumatic brain injury. Classical biomarkers including APP, α-spectrin, and neurofilaments are systematically reviewed. Besides, the study team speaks highly of the sodium channel in Ranvier nodes as a new biomarker for axonal injury. They propose to find more characterization of TAI to elucidate the progress of axonal injury, which may shed light on the intervention in different TAI phases. Here, I have a question. Line 270: Does the axonal bulb mean retraction ball?
Overall, the review is great and contains updated information on TAI.
Author Response
The paper reviews the phenotypes of diffuse axonal injury following traumatic brain injury. Classical biomarkers including APP, α-spectrin, and neurofilaments are systematically reviewed. Besides, the study team speaks highly of the sodium channel in Ranvier nodes as a new biomarker for axonal injury. They propose to find more characterization of TAI to elucidate the progress of axonal injury, which may shed light on the intervention in different TAI phases. Here, I have a question. Line 270: Does the axonal bulb mean retraction ball?
Following axonal disconnection, the subsequent retraction of the axonal segment and accumulation of axonally transported organelles and proteins forms a retraction ball (or bulb).
Here we are referring more generally to the accumulation of axonally transported material and subsequent axonal swelling as axonal bulbs, as standard light microscopy techniques cannot assess whether this is a terminal (or retraction) bulb relating to axonal disconnection or a potentially reversible form of pathology in an intact axon.